# Gram-Positive Bacteria-Like DNA Binding Machineries Involved in Replication Initiation and Termination Mechanisms of Mimivirus

**DOI:** 10.3390/v11030267

**Published:** 2019-03-17

**Authors:** Motohiro Akashi, Masaharu Takemura

**Affiliations:** Laboratory of Biology, Department of Liberal Arts, Faculty of Science, Tokyo University of Science, Shinjuku, Tokyo 162-8601, Japan; takemura@rs.kagu.tus.ac.jp

**Keywords:** DNA replication, *ori*, mitochondria, Rickettsia, gram-positive bacteria, APMV, Mimivirus, giant virus, eukaryogenesis

## Abstract

The detailed mechanisms of replication initiation, termination and segregation events were not yet known in *Acanthamoeba polyphaga mimivirus* (APMV). Here, we show detailed bioinformatics-based analyses of chromosomal replication in APMV from initiation to termination mediated by proteins bound to specific DNA sequences. Using GC/AT skew and coding sequence skew analysis, we estimated that the replication origin is located at 382 kb in the APMV genome. We performed homology-modeling analysis of the gamma domain of APMV-FtsK (DNA translocase coordinating chromosome segregation) related to FtsK-orienting polar sequences (KOPS) binding, suggesting that there was an insertion in the gamma domain which maintains the structure of the DNA binding motif. Furthermore, UvrD/Rep-like helicase in APMV was homologous to *Bacillus subtilis* AddA, while the chi-like quartet sequence 5′-CCGC-3′ was frequently found in the estimated *ori* region, suggesting that chromosomal replication of APMV is initiated via chi-like sequence recognition by UvrD/Rep-like helicase. Therefore, the replication initiation, termination and segregation of APMV are presumably mediated by DNA repair machineries derived from gram-positive bacteria. Moreover, the other frequently observed quartet sequence 5′-CGGC-3′ in the *ori* region was homologous to the mitochondrial signal sequence of replication initiation, while the comparison of quartet sequence composition in APMV/*Rickettsia*-genome showed significantly similar values, suggesting that APMV also conserves the mitochondrial replication system acquired from an ancestral genome of mitochondria during eukaryogenesis.

## 1. Introduction

Understanding the mechanism of genomic replication for all organisms, including the “giant viruses”, is an important scientific endeavor. Mimivirus, the first giant virus to be discovered, has a 750-nm-long virion and a 1.2 Mb linear dsDNA genome [1,2]. The method of replication termination for *Acanthamoeba polyphaga mimivirus* (APMV) has been previously hypothesized [3]. The first model suggested that the replication of the lagging strand of APMV’s linear genome is mediated by homologous recombination of approximately 617 bp located on both ends of the viral chromosome, similar to T4 phage replication, and is processed with Mimivirus R555 recombinase (Mre11/Rad50 fusion protein) [3,4]. Recently, the second model of replication termination and segregation of APMV was proposed [5]. In this model, the FtsK-like protein (also called packaging ATPase), binds FtsK orienting polar sequences (KOPS) and is localized to both ends of the nucleosome, resulting in chromosome segregation by the recombination of *dif* sequences [5]. The KOPS is the recognition sites of FtsK protein, and this protein controls the chromosome segregation in bacteria [6]. The second model reinforces the first model regarding homologous recombination between chromosomal ends. However, it should be noted that the bacteria do not perfectly conserve KOPS among species [6,7,8], suggesting that APMV could have KOPS of its own, which might be similar to bacterial KOPS. 

The initiation of DNA replication of bacteria is mainly driven by DnaA-DnaA box interaction, which subsequently unwinds dsDNA via DnaB [9]. Rep protein, related DNA helicase such as DnaB, initiates plasmid replication [10]. The Rep protein relates not only to plasmids but also to chromosome replication in *Escherichia coli* [11]. The homologues of these helicases in gram-positive bacteria are PcrA and AddAB; both are involved in DNA repair, and the former is involved in rolling-circle replication [12,13,14]. Especially, AddAB mediates the homologous recombination with recognizing the five-nucleotide sequence called chi [12], so the recombination is sequence specific. The sequence chi is known as a site-specific recombination site that is catalyzed by the RecBCD pathway in *E. coli*, and RecB is a homologous helicase to AddA [15,16]. Chi sequence varies among bacterial species: *E. coli*: 5′-GCTGGTGG-3′; *Bacillus subtilis*: 5′-AGCGG-3′; *Lactococcus lactis*: 5′-GCGCGTG-3′ [12,15,16,17]. Based on the model of replication termination in APMV [5], the DNA replication initiation mechanism might also be homologous to that of bacteria. If so, the pair of bacteria-like DNA sequences and their recognition proteins which are related to the DNA replication initiation and segregation could be found in APMV. Plotting the nucleotide composition bias called genomic GC skew is a tool for visualizing the bias of the nucleotide composition on the genome, which is able to determine the origin of replication since the values of GC skew switch across the replication origin and its terminus [18], and which has been determined in bacteria with some improvements of the technique [19,20,21]. This nucleotide composition bias shaping the genomic polarity is thought to have results of the mutation and selection pressure against the different replication mechanism of leading/lagging strand [22,23]. Although the origin of replication remains putative in Mimivirus, the GC skew analysis against the Mimivirus genome with high resolution may facilitate the detection of the signal sequence of DNA replication initiation. Therefore, using bioinformatics, we analyzed the APMV genome to determine the DNA replication initiation/termination segregation mechanism in detail, starting with the GC skew analysis.

## 2. Materials and Methods

### 2.1. GC/AT, Coding Sequence (CDS) Skew Analysis

GC and AT skew of the APMV genome (AY653733.1) was analyzed using a method described previously [20,21]. Each index was calculated using the following formulas: GC skew = [G-C]/[G+C]; AT skew = [A-T]/[A+T] (window size: 10,001 bp; step size: 1000 bp), and the GC/AT skew and cumulative graph of these were plotted. To calculate coding sequence (CDS) skew, we indexed the CDS direction on the APMV genome (direction of the gene (D): positive: +1; negative: -1) and the CDS length (L). Subsequently, the CDS skew index was calculated against every CDS using the following formula: CDS skew = [D] × [L] (Figure 2b). The CDS skew and cumulative CDS skew were plotted with the CDS start positions.

### 2.2. Correlation Analysis of the CDS Length of Left/Right Side from the Estimated Ori Region

The CDS length of the left or right sides from the estimated origin (380,698 bp), and the CDS length of the positive or negative direction against the APMV genome (AY653733.1), were plotted using a box plot. The statistical differences between each of two groups were calculated by two-sample Kolmogorov–Smirnov (KS) test using “lawstat package” of R software (https://www.r-project.org) with default options.

### 2.3. Paralogous Gene Localization Analysis

We confirmed the three kinds of paralogous gene locations found on the two GC skew shift points (296,000 bp and 882,000 bp): *ankyrin repeat, serine/threonine protein kinase*, and *collagen triple helix repeat containing protein*. First, we made a gene list of the CDS information annotated as “ankyrin containing protein”, “serine/threonine protein kinase”, or “collagen triple helix repeat containing protein” in the APMV genome (AY653733.1), containing the data of locus tags, start/end physical positions, gene length and gene direction and GC content (Appendix A). Every pair of each of the three genes was listed and the distance between these pairs from the start position was calculated. This pair of directions on the physical position of the APMV genome was plotted using the Circos v0.69-6 software [24]. The pair of gene direction was also labeled (matched direction: forward-forward/reverse-reverse; mismatched direction: forward-reverse/reverse-forward). When drawing the graph with Circos, color coding was used to display the three kinds of genes, and states of matched/mismatched gene directions were displayed with shades of colors (ankyrin containing protein: blue; serine/threonine protein kinase: green; collagen triple helix repeat containing protein: red; matched: light; mismatched: shade). The columns of id, gene1_start_pos, gene2_start_pos, line_colour, and line_thickness in the Appendix A without header (Appendix A; the pane “Paralogous gene direction”) can be used for loading the data to Circos. The distance between every two CDSs in each of the three genes was plotted by the CDS direction-matching patterns on the APMV genome (“match” or “mismatch”). The statistical differences between the two groups were calculated by two-sample KS test using the “lawstat package” in the R software with default options.

### 2.4. Sequence Alignment Analysis Neighbor-Net Network Analysis

The accessions of analyzed FtsK with positions of motor (ATPase) domain and gamma domain were as follows: APMV: AAV50705.1, 5-215 aa, 216-284 aa; *E. coli*: NP_415410.1, 868-1242 aa, 1268-1324 aa; *L. lactis*: NP_267812.1, 312-668 aa, 695-749 aa; *Pseudomonas aeruginosa*: Q9I0M3, 343-716 aa, 749-803 aa; *B. subtilis*: WP_003231869.1, 346-703 aa, 582-723 aa. The accession numbers of analyzed UvrD/Rep-like helicase were as follows: APMV: AKI80299.1; *E. coli*: YP_026251.1 (Rep), AAA67609.1 (UvrD), NP_417297.1 (RecB); *L. lactis*: WP_003132060.1 (PcrA), WP_010905024.1 (AddA); *B. subtilis*: WP_003233919.1 (PcrA), WP_003233100.1 (AddA). The dataset without the APMV sequence was used for alignment. Poor-quality sequences were masked using Prequal software [25], and sequence alignment analyses was performed using the MAFFT v7.222 software with “—auto” option [26]. Subsequently, the APMV sequence was added and realigned with MAFFT. To determine the alignment of the chi-binding site of AddA with APMV, two sequences of AddA from *L. lactis* and *B. subtilis* (WP_010905024.1 and WP_003233100.1) were recursively added with the APMV sequence. For the Neighbor-Net network analysis, alignment data containing APMV sequence were trimmed using trimAl 1.2rev59 software with “-strictplus” option [27], and the numbers of aligned residues used were: FtsK ATPase domain: 322 aa; FtsK gamma domain: 49 aa; UvrD/Rep-like helicase: 483 aa. The Neighbor-Net network tree was drawn by SplitTree4 (version 4.14.8) with 1000 bootstrap replicates [28,29].

### 2.5. KOPS Distribution in the Genome

The bacterial KOPS distributions on APMV and the bacterial genomes were plotted using the following genome data: APMV: AY653733.1; *L. lactis* Il1403: NC_002662.1; *E. coli* MG1655: NC_000913.3; *B. subtilis* 168: NC_000964.3. The information on the *ori* and *ter* positions of these bacteria was provided by Genome projector website (http://www.g-language.org/g3/) [30]. KOPS of each bacteria were listed as follows: *L. lactis*: 5′-GAGAAG-3′; *B. subtilis*: 5′-GAGAAGGG-3′; *E. coli*: 5′-GGGNAGGG-3′ [6,7,8].

### 2.6. Quartet Sequence Composition Analysis

Every quartet sequence compositions on APMV and the bacterial genomes were confirmed with the compseq program of the EMBOSS 6.6.0.0 software [31] with the “-word 4 ” option, using the following genome data: APMV: AY653733.1; *L. lactis* Il1403: NC_002662.1; *E. coli* MG1655: NC_000913.3; *B. subtilis* 168: NC_000964.3; *Rickettsia prowazekii*: NC_000963.1; *Homo sapiens* mitochondria (MT): CM001971.1. To compare the compositions between the estimated *ori* region and the whole genome of APMV, sequence composition was also confirmed on the 375–385 kb region, and the ratio of *ori* region per whole genome (quartet nucleotide composition ratio) was calculated as follows: quartet nucleotide composition ratio = [*ori* observed/expected frequency]/[all genome observed/expected frequency]. The Grubbs test was performed on this data to detect the highly observed sequence on the *ori* region with the “outliers” package of the R software (options: type = 11; opposite = TRUE; two.sided = TRUE). The mutual observed frequency of quartet sequence among APMV, APMV *ori* region, and bacteria were plotted, while statistical differences were confirmed by two-sample KS test using the “lawstat package” of the R software with default options.

### 2.7. Homology Modeling Analysis

Homology modeling analyses of the gamma domain of packaging ATPase (FtsK-like protein, 216-284 aa region of AAV50705.1) were performed using the I-TASSER server [32]. Template structures of *P. aeruginosa* (2J5O and 2VE9) were used for modeling [33,34]. 

## 3. Results

### 3.1. GC/AT Skew and CDS Skew Analyses

The GC skew plot showed the two highest and lowest peaks at the symmetrical position point of the genome (296 kb and 882 kb), and the plot was able to separate the three regions by these positions (Figure 1a). Both end regions (<296 kb, >882 kb) were increasing; however, the former value was negative and the latter value was positive, indicating that the 5‘ end contained a C nucleotide rather than G nucleotide, and the 3′ end contained a G nucleotide rather than a C nucleotide. The middle region of the graph (from 296 kb to 882 kb) was almost flat (Figure 1b), and the cumulative GC skew plot of this region was increasing (Figure 1b), indicating that the number of G and C nucleotides in this region was slightly skewed to the G nucleotide. The AT skew graph showed the shift point of the value, which corresponded to the peak of the valley of the cumulative AT skew graph at 382,000 ± 5000 bp region, suggesting that the origin of DNA replication is located in this region (Figure 1). However, the cumulative AC skew did not show any peaks (Appendix A).

CDS skew analysis showed that the shift point and cumulative CDS skew analysis exhibited a V-shaped graph, similar to the AT skew/cumulative AT skew (Figure 2a). Furthermore, the valley of this cumulative CDS skew graph was located at 382,698 bp, and the gene direction faced outward from this peak. Therefore, we concluded that the estimated *ori* region is located at the 382 kb position of APMV genome. The location of the estimated *ori* region is biased toward the 5′ end from the center of the genome; however, there were no significant differences in CDS length between the right and left sides of the estimated *ori* region and between the positive and negative strands (Appendix A). We found the same paralogous genes on the GC skew shift points (296,000 bp, 882,000 bp): ankyrin repeat, serine/threonine protein kinase, and collagen triple helix repeat containing protein. Paralogues of these genes were located on the line of symmetry position in the genome, especially *collagen triple helix repeat containing protein* gene, which exhibited the exact positions of the GC skew shift points with opposite gene direction, suggesting that the nucleotide compositions of these paralogues formed the shift points (Figure 3a). Additionally, we analyzed the gene-to-gene distances between each pair of paralogues for each of the three genes, indicating that the distance between the gene direction matched pair was shorter than that between the mismatched pair (fold change: 2.6, *p* < 0.05, Figure 3b). Thus, APMV forms the double-folding structure and is the cause of paralogue generation by homologous recombination (Appendix A).

### 3.2. Initiation of DNA Replication

#### 3.2.1. Sequence Analysis of UvrD/Rep-Like Helicase

Since the replication initiation mechanism had not yet been analyzed in APMV, we sought to determine the protein that participated in the initiation of DNA replication. In doing so, we discovered a similar sequence to Rep helicase that was related to the both chromosome and plasmid replication [10,35]. This protein has already been annotated as “UvrD/REP helicase family protein”, suggesting that this protein is possibly an initiator of DNA replication in APMV (accession ID of NCBI protein database: AKI80299.1). We aligned the UvrD/REP helicase family protein of APMV (AKI80299.1) with the bacterial homologues: Rep, UvrD, and RecB of *E. coli*; PcrA and AddA of *B. subtilis*; and *L. lactis* [10,12,13]. The alignment and phylogenetic analysis showed that the UvrD/Rep-like helicase of APMV is a close relative to the AddA rather than Rep (Figure 4a,c). Seven regions that were known to be conserved among AddA and other helicases [36] were also highly conserved in the UvrD/Rep-like helicase (Figure 4a). AddA forms a heterodimer with AddB, which recognizes the chi sequence of *B. subtilis* [37]. Therefore, we compared the homologous region of the chi binding site of AddA with UvrD/Rep-like helicase of APMV. The results showed that there were three regions containing chi binding sites in UvrD/Rep-like helicase of APMV, and two out of seven sites were conserved: Q1115 and I1157 of *Bacillus*-AddA corresponded to the Q922 and I924 of UvrD/Rep-like helicase of APMV; and Y1204 of *Bacillus*-AddA presumably corresponded to the Y973 of UvrD/Rep-like helicase of APMV, which was shifted one amino acid residue to the C-terminus from the homologous site against Y1204 in *B. subtilis* (Figure 4b). Although we have aligned the other region containing four DNA-binding residues between the 1012–1019 aa region of *Bacillus*-AddA with the 610–617 aa region of UvrD/Rep-like helicase, both of which are constructed with polar amino acids, we could not find any conserved sequences of DNA binding sites (K1013, S1015, V1016, and S1017 in *Bacillus* AddA, Figure 4b). These four residues in *Bacillus*-AddA bind the phosphate at the 3′ end of the chi sequence [37], so that the content of polar amino acids in this region, rather than the exact amino acid sequence, is important for *Bacillus*-AddA to bind to the chi sequence. Therefore, we estimated that the 610–617 aa region of UvrD/Rep-like helicase of APMV could also bind the 3′ end of the chi-like sequence in the APMV genome. Altogether, the UvrD/Rep-like helicase of APMV was similar to the AddA of gram-positive bacteria, and thus this protein would presumably recognize the chi sequence.

#### 3.2.2. Signal Sequence of the Initiation of DNA Replication

Subsequently, we sought to find the sequence that was related to the origin of recognition on the probable *ori* region. To accomplish this, we first calculated the quartet nucleotide composition ratio of the *ori*-containing region per total genome (Figure 5). For details of the calculation of the ratio, see Materials and Methods. The four types of sequences that had a ratio greater than 2 contained 75–100% of GC nucleotides, and two of four sequences, CGGC and CCGC, exhibited significantly different ratios from the other sequences (*p* < 0.05, Figure 5a). Furthermore, two sequences (CCGC and GCGG) were complementary to each other (Figure 5a). This pair of sequences was identical to four out of five nucleotides on the chi sequence, which was recognized by *B. subtilis*-AddA (5′-AGCGG-3′) [12]. Additionally, the sequence CGGC is known to be a part of a complementary sequence that is known as a replication signal sequence of mitochondrial genome in humans, 3′-GGCCG-5′ [38]. Therefore, we analyzed the distribution of these quartet sequence in the APMV genome. The sequence densities of CCGC and GCGG in the APMV genome showed that both sequences had two peaks, and one of each peak was located on the *ori*-containing region, while the others were on the axial symmetric position (about 800,000 bp, Figure 5b). Similar to this, those of CGGC and GCCG exhibited two peaks and the peak on the 5′ side in GCCG located on the *ori*-containing region, although that of the CGGC slightly skewed from the *ori*-containing region to the 5′ end of the genome (Figure 5c). Each type of quartet sequence composition frequencies was calculated in APMV, four kinds of bacteria (*B. subtilis*, *L. lactis* and *E. coli*, *and R. prowazekii*), and human mitochondrial genome (*H. sapiens* MT). *R. prowazekii* is considered as an ancestor of mitochondria [39], while the mitogenome sequence similarity against the Mimivirus genome has been reported recently [40]. These frequencies were then plotted in every pair of bacteria, mitochondria, APMV, and the *ori*-containing region of APMV (375–385k) (Figure 6). Every pair without APMV-APMV-*ori* and APMV-*R.prowazekii* exhibited a significant difference of sequence composition (*p*<0.05, Figure 6). *R. prowazekii* possesses the most similar sequence composition compared with APMV (*p* = 0.2528). The composition similarity of the *R. prowazekii*-APMV pair was higher than those of the *R. prowazekii*-*H. sapiens* MT and APMV-*H. sapiens* MT pairs (*R. prowazekii*-*H. sapiens* MT: *p* = 0.04685; APMV-*H. sapiens* MT: *p* = 0.00146), suggesting that the APMV conserves the sequence derived from an ancestor of mitochondria rather than from highly evolved mitochondria in humans. Interestingly, the composition of human mitochondria and *L. lactis* also showed a high similarity (*p* = 0.3552) rather than the *R. prowazekii*-*H. sapiens* MT pair, suggesting that the mitochondrial genome still conserves a remnant of bacterial characteristics.

### 3.3. Termination of DNA Replication and Chromosome Segregation

#### 3.3.1. Sequence Analysis of FtsK-Like Protein in APMV

FtsK mainly conserves the motor domain (alpha/beta domain) and gamma domain, the latter of which is our main target, and recognizes KOPS [5,26]. The DNA binding sites of FtsK has been determined in *P. aeruginosa* [34,41]. We aligned the gamma domain of APMV with *P. aeruginosa* and other bacteria (*L. lactis*, *B. subtilis,* and *E. coli*), revealing that the DNA binding sites determined in *P. aeruginosa* were conserved to almost the same degree among APMV and bacteria. The phylogenetic analysis of FtsK-gamma domain among these species indicated that the sequence of *L. lactis* was most closely related to the APMV-FtsK gamma domain (Figure 7b). However, in APMV, there was a 10 aa insertion in the region where the DNA binding sites were localized (Figure 7a), suggesting that DNA-binding activities of APMV-FtsK had possibly collapsed. Therefore, we confirmed the three-dimensional structure via homology modeling of the gamma domain in APMV-FtsK, using two models from *P. aeruginosa* as templates [34,41]. As a result, the DNA binding sites of APMV-FtsK were thought to be topologically reconstructed and conserved by this insertion (Figure 8). Estimated DNA binding sites against the DNA backbone were K242, K243, K253, and K256, which corresponded with R770, K771, R778, and R781 in *P. aeruginosa,* respectively, and the binding site for KOPS specifically was N246 at APMV-FtsK, which corresponded to N777 in *P. aeruginosa*. K242, K243, and N246 were placed on the insertion (Figure 7a, Figure 8a) [33]. These four basic amino acids localize in two helix motifs, which bind to the KOPS region and support the recognition of KOPS by asparagine between the two helix motifs (Figure 8a,b) [33]. Next, we confirmed the conservation of the ATP binding sites on the FtsK in APMV. The FtsK conserves the Walker A motif (the ATPase active site), and in *P. aeruginosa*, substitution of amino acid residue in this motif leads to the deactivation of ATPase (K472N) [41]. The sequence alignment showed that the motor domain of APMV conserved lysine on the Walker A motif (K32) as well as other three bacteria (Figure 9a). The other ATP binding sites found in *P. aeruginosa* [41] were partially conserved in the APMV-FtsK (Figure 9a). However, some amino acids were not conserved, even in bacteria (R418 and H675 of *P. aeruginosa*-FtsK, Figure 9a), suggesting that these amino acid residues in binding sites could be replaced with other amino acid residues. Phylogenetic analyses of these motor domains indicated that the sequence of APMV was distinct from bacteria (Figure 9b). Therefore, APMV-FtsK might have speciated from its bacterial group prior to the emergence of bacterial FtsK.

#### 3.3.2. Distribution Pattern of KOPS on APMV Genome

The most frequently observed type of KOPS sequence was from *L. lactis*, whereas those of *B. subtilis* and *E. coli* were hardly encountered (*L. lactis*, 5′-GAGAAG-3′: 225; *B. subtilis*, 5′-GAGAAGGG-3′: 5; *E. coli*, 5′-GGGNAGGG-3′: 9; Figure 10a). This KOPS distribution pattern was different between the positive and negative strand, and the density graphs showed that these two patterns crossed at the estimated *ori* position (Figure 10b). These KOPS distribution patterns were analyzed in each bacterial genome, and the results suggested that the density of KOPS on the positive/negative strand switched on the exact points of the *ori*/*ter* region (Figure 10c). This is similar to the distribution pattern of *L. lactis*-derived KOPS in the APMV genome and to KOPS in the bacterial genomes, indicating that *L. lactis* KOPS could be one of the commonly used termination sequences in DNA replication in APMV. It should be noted that we did not have evidence that the other two KOPS, 5′-GAGAAGGG-3′ and 5′-GGGNAGGG-3′, found in APMV, were inactive (Figure 10a).

## 4. Discussion 

The GC/AT skew bias is the result of replication bias in bacteria [19], and the replication-transcription conflict causes a high mutation rate in genes, causing genetic transcription and DNA replication to be co-directional [42]. We determined the estimated *ori* region using high-resolution cumulative skew graphs of AT nucleotides and CDSs (382 kb, Figure 1 and Figure 2), which suggests that the transcription and DNA replication of the APMV genome are co-directional. This characteristic is not unique to cellular organisms but is the same in APMV. Both bacteria and viruses replicate faster than eukaryotes. APMV has a 1.2 Mb genome, a size which is similar to that of bacteria. Therefore, its large genomic structure would likely be constructed while facing the selection pressure of DNA replication.

The negative and positive values on both ends of the GC skew plot suggest that the G/C nucleotide composition switches between the 5′ region (<296 kb) and the 3′ region (>882 kb). Both end regions have *dif* sequences, which have been previously estimated [5], suggesting that this nucleotide composition bias is perhaps a key factor for replication termination with directional homologous recombination (Appendix A). Indeed, both regions harbor paralogous genes of ankyrin repeats in opposite directions, indicating that recombination between the 5′ and 3′ ends frequently occurs (Figure 3a). Moreover, the ankyrin repeats are symmetrically placed between *dif* 2 and *dif* 3; thus, these two *dif* sequences might often be used for termination and segregation (Figure 3a). A model for the termination of DNA replication in APMV has been previously described, hypothesizing that the genomic DNA bent symmetrically during DNA replication of the 3′ end [3]. The symmetrical distribution of the sequences and genes in APMV suggest that homologous recombination and/or sequence insertions would occur prior to chromosome segregation. Furthermore, the large, symmetrical inversion between APMV and *Megavirus chillensis* (lineage C of *Mimiviridae*) [43] indicates that the topology of the genomic DNA during replication led to the diversification of the viral family *Mimiviridae*.

In bacteria, the *dnaA* box sequence on the *ori* region is recognized by DnaA when DNA replication is initiated [9], and the Rep protein is used when plasmid replication is initiated [44]. Rep is also involved in chromosomal replication, and a lack of Rep function can cause delay in chromosomal replication [11]. In *E. coli*, RecB, which is a homologous helicase to Rep, acts in the reconstruction of a stalled replication fork in the RecBCD pathway, while recognizing chi sequences (5′-GCTGGTGG-3′) [15,16]. The gram-positive bacteria *B. subtilis* encodes the homolog of the RecB, AddA, and recognizes the chi sequence 5′-AGCGG-3′ [12]. Based on our results, UvrD/Rep-like helicase of APMV is more similar to AddA than it is to Rep, and a portion of the gram-positive chi-like sequence (5′-CCGC-3′) is frequently found in the estimated *ori* region of the APMV genome (Figure 4 and Figure 5) This suggests that the initiation of DNA replication is presumably mediated by the interaction between the UvrD/Rep-like helicase and chi-like sequences in APMV. In *B. subtilis,* AddA, Q1155, and I1157, homologous residues of Q922 and I924 in UvrD/Rep-like helicase of APMV are known to recognize the fourth and fifth G nucleotides of the chi sequence (5′-AGCGG-3′) [37]. Therefore, these conservations are appropriate for binding APMV helicase to the GCGG sequence, of which the complementary sequence is frequently found in the estimated *ori* region of the APMV genome (Figure 4b and Figure 5). However, S1015, V1016, S1017, and Y1204 in *B. subtilis* AddA bind phosphate at the 3′ end of the chi sequence [37], suggesting that these residues are not involved in the recognition of a specific chi sequence. Phosphate binding sites were not conserved in UvrD/Rep-like helicase of APMV at the sequence level, however these regions contain polar amino acids similar to *B. subtilis* AddA (Figure 4b), which indicates that the UvrD/Rep-like helicase of APMV could also bind phosphate at the 3′ end of the chi sequence. Furthermore, the other quartet sequence, 5′-CGGC-3′, which was detected on the estimated *ori* region of APMV (Figure 5a,c), is reported to be a part of the DNA replication signal sequence of mitochondria (3′-GGCCG-5′) [38], indicating that the DNA replication of APMV is also initiated by host replication machineries involved in the replication of mitochondrial DNA.

KOPS and FtsK determine the region of DNA replication termination, and FtsK mediates DNA segregation process in bacteria [7,8]. KOPS are different among bacteria [6,7,8], in that the KOPS recognition mechanism is thought to be defined by the structure of FtsK and the sequence of KOPS itself. Our results showed that, in the APMV genome, the distribution pattern of *L. lactis*-KOPS was similar to those of bacteria (Figure 10b,c). The sequence analysis of the FtsK gamma domain, which interacts with KOPS, also showed a high similarity between *L. Lactis* and APMV (Figure 7), suggesting that the pair of KOPS and the FtsK-gamma domain structure in APMV might be homologous to that of *L. lactis*. Furthermore, we found that there was an insertion in the gamma domain of APMV-FtsK. Interestingly, the three-dimensional structure estimated by homology modeling revealed that this insertion was reconstructed and conserved in the DNA-binding domain (Figure 8). We also showed that the Walker A motif was conserved in the FtsK-motor domain of APMV (Figure 9a). These results suggest that APMV-FtsK might be functionally homologous to that of bacteria. The phylogeny of APMV-FtsK (gamma domain) was also found to be similar to that of *L. lactis* (Figure 7b). Therefore, the mechanisms of interaction between these proteins and specific sequences are presumably homologous between APMV and bacteria such as *L. lactis*, although further molecular biological studies and structural analyses are required to certify this model.

The phylogenetic relations between *Rickettsia* and mitochondria and between mitochondria and Mimivirus were described previously [39,40]. Interestingly, the genome size and GC content of *Rickettsia* are 1.1 Mb and 29%, respectively [39], which is highly similar to APMV (1.2Mb, 28%). Furthermore, the backbone of the Mimivirus genome is reported to be derived from the ancestor of mitochondria [40]. Considering the quartet sequence similarities (Figure 6) and the phylogenetic relation between *Rickettsia* and mitochondria, the ancestor of Mimivirus infected the ancestor of eukaryotic cells (last archaeal common ancestor, LACA) before the endosymbiosis of mitochondrial ancestor to the first eukaryotic common ancestor (FECA), while the ancestor of mitochondria and *Rickettsia* also infected to the ancestor of eukaryotic cells. This ancestral virus presumably harbored shorter genomic DNA than the present-day Mimivirus, and it acquired the long genome from ancestor of mitochondria by genomic fusion. The sequence 5′-CGGC-3′ found in the APMV *ori* region might also have been acquired from the ancestor of mitochondria, and is still conserved as a DNA replication signal sequence (3′-GGCCG-5′) in the mitochondrial genome [38]. It has been reported that the comparison between mitochondrial genes and the *Rickettsia* genome shows much higher similarities than that between mitochondrial genes and megaviral genome (poxvirus), while the synteny of these three species are significantly conserved [45]. Thus, according to our results, the genomic remnants of the ancestor of mitochondria may be still conserved in the Mimivirus genome to a greater extent than in the poxvirus genome. Moreover, the co-infection (or preying on) of the LACA cells might have occurred not only in ancestor of mitochondria and Mimivirus but also in other bacteria, and therefore APMV conserves the bacteria-like machineries such as UvrD/Rep-like helicase and APMV-FtsK derived from the co-infected ancestor of gram-positive bacteria by horizontal gene transfer.

## 5. Conclusions

Here, we presented a proposed model of the initiation and termination of DNA replication and chromosome segregation for APMV. The estimated *ori* region exists at the 382 kb position in the genome, which contains the chi-like sequence recognized by *B. subtilis* AddA, which is homologous to the UvrD/Rep-like helicase of APMV. The other sequence has a homology of a DNA replication signal sequence of mitochondria, indicating that the DNA replication of APMV may initiate with the replication machineries of mitochondria. The KOPS distribution pattern and the structure of APMV-FtsK indicate that the KOPS recognition system by APMV-FtsK is similar to that of *L. lactis*. Consequently, replication initiation, termination and segregation systems of APMV are presumably mediated by DNA repair machineries, similar to that of gram-positive bacteria, such as *L. lactis*. Furthermore, the comparison of quartet sequence compositions shows the similarity between APMV and *Rickettsia,* which may have the closest common ancestor of mitochondria, indicating that Mimivirus has acquired a large bacteria-like genome and its DNA replication machineries from ancestor of mitochondria during the co-infection to the LACA cells. The evolutionary history of APMV remains unclear; however, the further analyses of such a chimeric genome of APMV may illustrate the early stage of evolution of eukaryotic cells and Mimivirus.

## Figures and Tables

**Figure 1 viruses-11-00267-f001:**
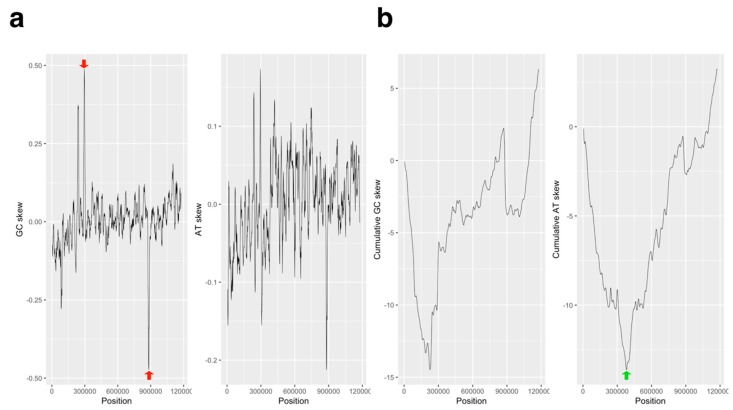
Analyses of the GC/AT skew in the *Acanthamoeba polyphaga mimivirus* (APMV) genome. (**a**) GC and AT skew plot of APMV genome (AY653733.1, step size: 1000 bp; window size: 10,001 bp). Red arrows on the GC skew plot indicate the highest/lowest peaks, which are located at the symmetrical position of the genome (GC skew: 0.488234 at 296,000 bp; -0.472989 at 882,000 bp). (**b**) Cumulative GC and AT skew plots corresponding to the graphs on the panel (a). The green arrow on the graph on the cumulative AT skew plot shows the lowest valley point on this graph, which was estimated in the genomic region as an origin of replication (382,000 ± 5000 bp).

**Figure 2 viruses-11-00267-f002:**
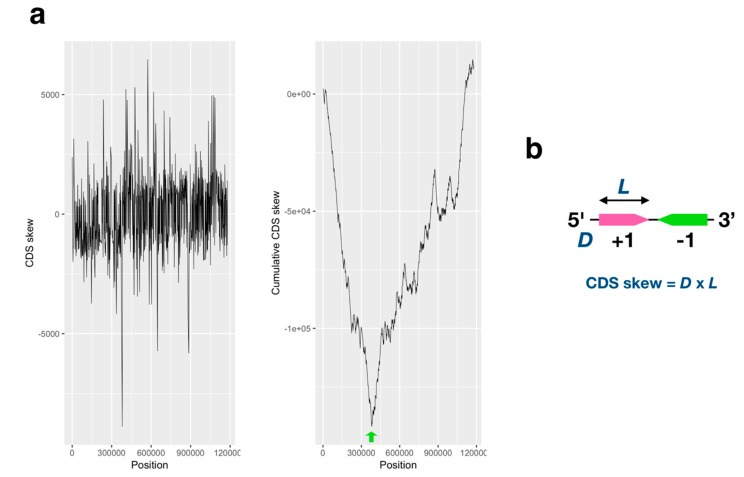
Analyses of the coding sequence (CDS) skew of the APMV genome. (**a**) CDS skew plot (left) and cumulative CDS skew plot (right) of the APMV genome (AY653733.1). The green arrow on the graph on the cumulative CDS skew plot shows the lowest valley point, which was estimated to be genomic region of the origin of 382,698 bp. (**b**) Calculation of the CDS skew index. D: Direction values of the CDS against the APMV genome (AY653733.1); L: CDS length (bp). Pink and green arrows indicate the positive and negative CDS direction on the APMV genome, respectively.

**Figure 3 viruses-11-00267-f003:**
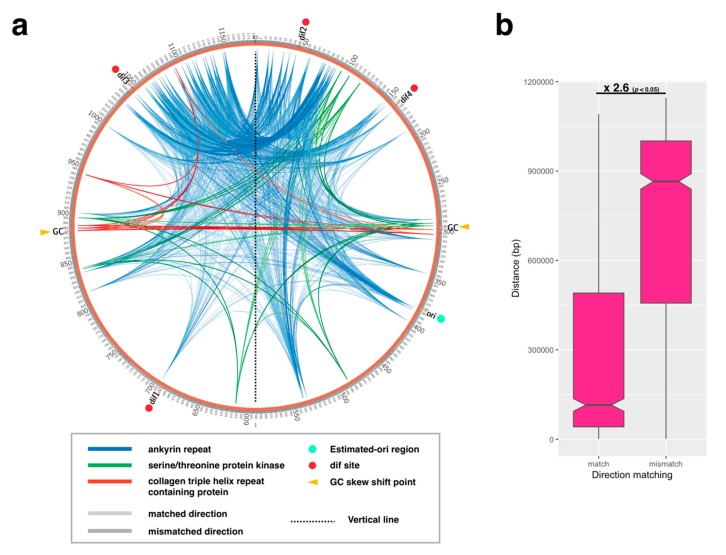
Paralogous gene localization found at the GC skew shift point. (**a**) The three genes found on the GC skew shift point were plotted (ankyrin repeat, serine/threonine kinase, and collagen triple helix repeat containing protein). Four *dif* sites (*dif*1-4) were described previously (see text). Each of the three genes were separately categorized as matching or mismatching the coding sequence (CDS) direction on the *Acanthamoeba polyphaga mimivirus* (APMV) genome (AY653733.1) with shades of colors (light: matched direction; dark: mismatched direction). (**b**) Distances between each pair of CDSs in each of the three genes were plotted by the CDS direction matching patterns on the APMV genome (“match” or “mismatch”). Labels above the box plots indicate the fold change between the two values with the resulting *p*-value from KS-tests.

**Figure 4 viruses-11-00267-f004:**
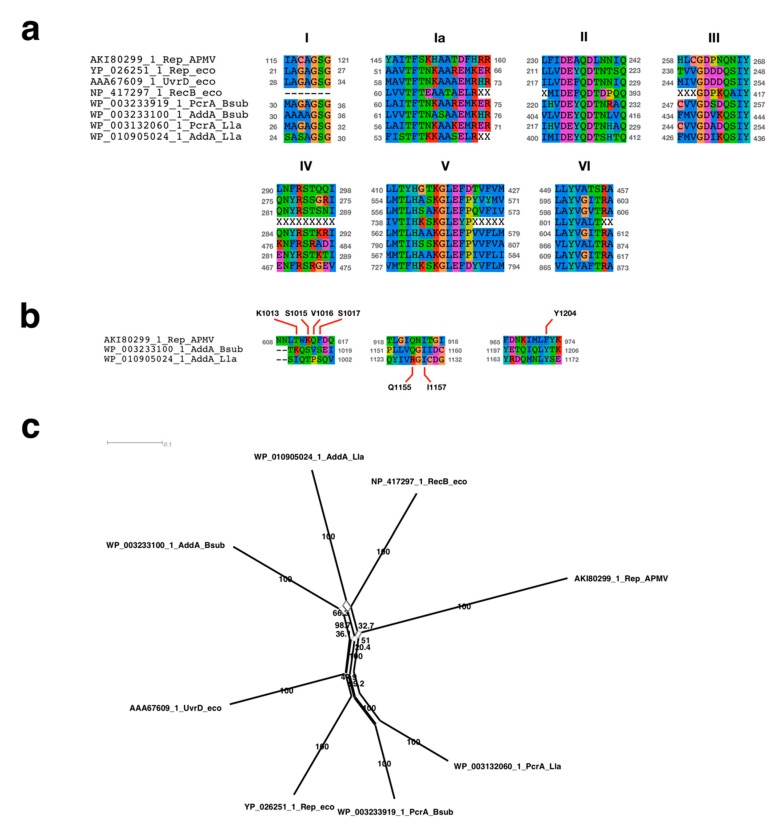
Sequence analyses of UvrD/Rep-like helicase of APMV. (**a**) Sequence alignment of seven conserved regions among APMV and three bacteria. Character “X” on the alignment indicates the residues masked by the prequel program [25]. Accession number and species name are labeled on the left side of the alignments. Grey colored numbers, which are located on both sides of the alignments, indicate the physical positions of each sequence. (**b**) Sequence alignment of chi sequence binding sites. Accession number and species name are labeled on the left side of the alignment. Grey colored numbers, which are located on both sides of the alignments, indicate the physical positions of each sequence. Labels above/under the alignment indicate the amino acids related to the chi sequence binding sites determined using *B. subtilis*-AddA [37]. (**c**) Neighbor-Net network tree of the UvrD/Rep-like helicase. Accession number and species name corresponding to the sequence alignment are labeled at the end of the branch. Numbers on nearby branches indicate bootstrap test values with 1000 replicates. Scale bar: number of substitutions per site.

**Figure 5 viruses-11-00267-f005:**
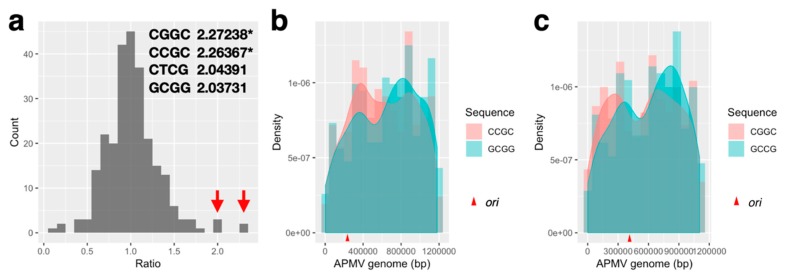
Comparative analysis of the quartet nucleotide composition ratio between the *ori*-containing region and the total genome of APMV. (**a**) Frequency distribution of fold-changes (= [*ori*-containing region]/[whole genome], *ori*-containing region: 375–385 kb of AY653733.1). Red arrows indicate that the fold-changes of the bins are greater than two. Sequences and fold-changes greater than two are listed on the top right of the figure. Asterisks indicate significant outliers from the population calculated by Grubbs test (*p*<0.05). (**b**,**c**) Density graph of the 5′-CCGC-3′/5′-GCGG-3′ and 5′-CGGC-3′/5′-GCCG-3′ on the APMV genome (AY653733.1). Red arrowhead on the scale indicate the estimated location of the *ori* region (382,000 ± 5000 bp).

**Figure 6 viruses-11-00267-f006:**
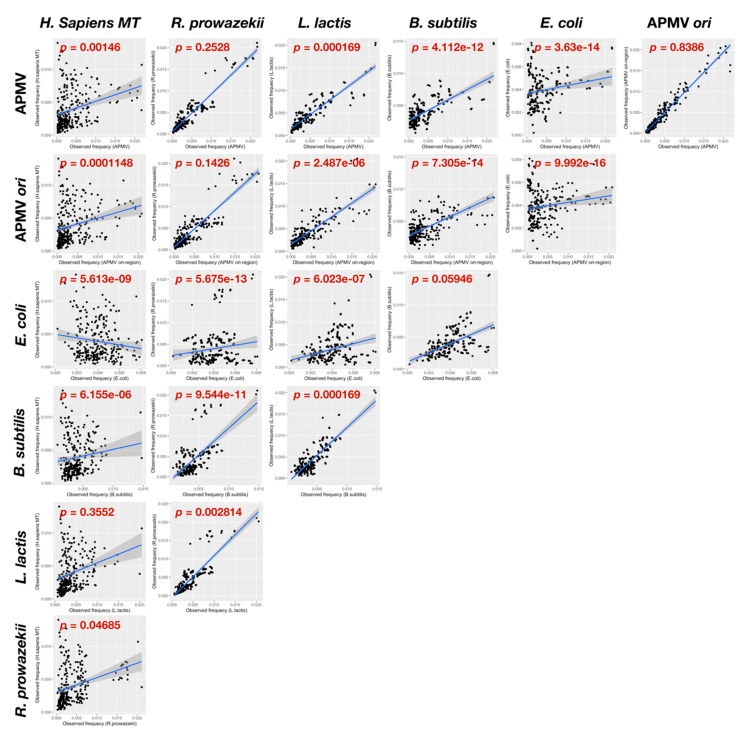
Comparative analysis of quartet nucleotide composition frequencies among APMV, human mitochondria (*H. sapiens* MT), and four bacterial genomes (*Escherichia Coli*, *Bacillus subtilis, Lactococcus lactis, Rickettsia prowazekii)*. Pairs of every frequency between two species were plotted with an approximate line. “APMV *ori*” indicates the frequencies of sequence compositions in the estimated location of the *ori* region (375–385 kb region of AY653733.1). *p*-values in each top left corner of the graphs are the significant differences calculated between two groups as determined by KS test.

**Figure 7 viruses-11-00267-f007:**
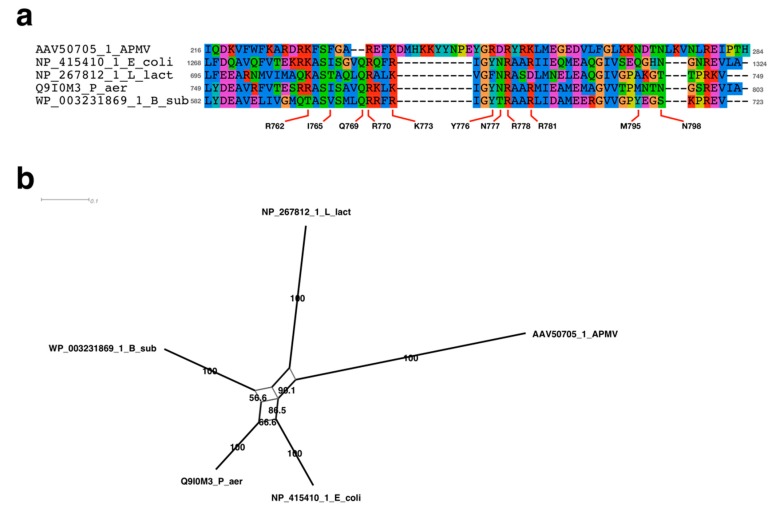
Sequence analyses of APMV-FtsK gamma domain. (**a**) Sequence alignment of FtsK-like protein gamma domain. Accession number and species name are labeled on the left side of each alignment. Grey numbers located on both sides of the alignments indicate the exact positions of each sequence. Amino acids labeled under the alignment indicate the amino acid residues responsible for DNA binding sites of *Pseudomonas aeruginosa* [25]. (**b**) Neighbor-Net network tree of FtsK gamma domains. Accession number and species name corresponding to the sequence alignment are labeled at the end of the branch. Numbers on nearby branches indicate bootstrap test values with 1000 replicates. Scale bar: number of substitutions per site.

**Figure 8 viruses-11-00267-f008:**
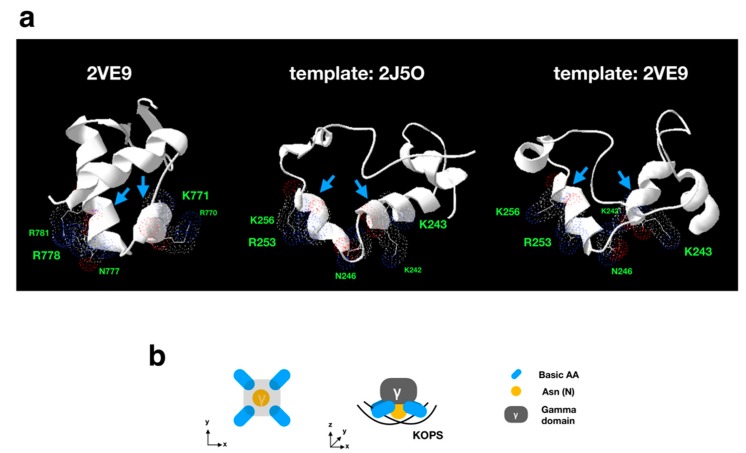
Homology modeling analysis of APMV-FtsK gamma domain. (**a**) Estimated structures of FtsK gamma domain using two different templates of *P. aeruginosa*-FtsK (PDBID; middle: 2J5O; right: 2VE9). The left model is the gamma domain of *P. aeruginosa*-FtsK (PDBID: 2VE9). Side chains on the left model are directly bound to FtsK orienting polar sequences (KOPS), while the other two models are estimated amino acids, which are functionally homologous to 2VE9. Blue arrows indicate two helix motifs harboring KOPS binding residues. (**b**) Model of the gamma domain of APMV-FtsK against KOPS. Four basic amino acids and Asn (N) correspond to the side chain in every model on panel (a).

**Figure 9 viruses-11-00267-f009:**
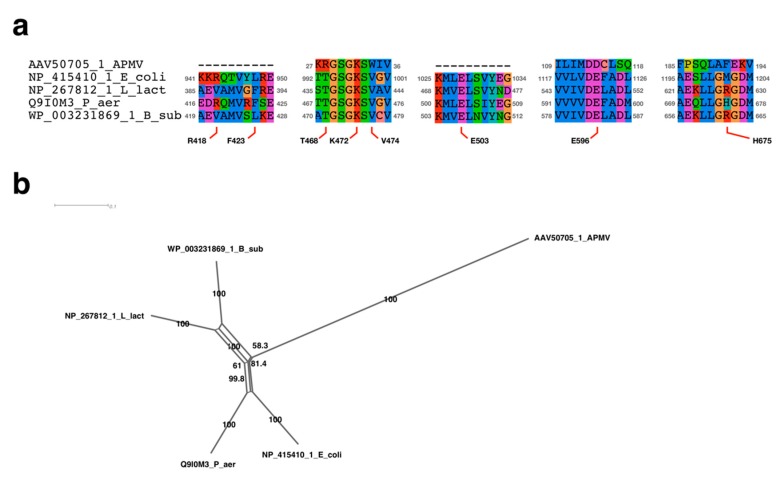
Sequence analyses of APMV-FtsK ATPase domain. (**a**) Motor domain of FtsK-like protein sequence alignment. Accession number and species name are labeled on the left side of alignments. Grey numbers located on both sides of the alignments indicate the exact positions of each sequence. Labels under the alignment indicate the amino acids related to ATPase activities determined in *P. aeruginosa* [41]. (**b**) Neighbor-Net network tree of the gamma domains. Accession number and species name correspond to the sequence alignment labeled at the end of the branch. Numbers on nearby branches indicate bootstrap test values with 1000 replicates. Scale bar: number of substitutions per site.

**Figure 10 viruses-11-00267-f010:**
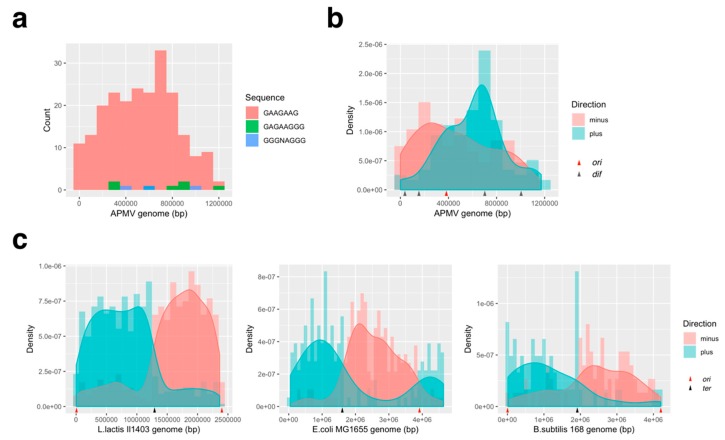
Bacterial KOPS distribution on APMV. (**a**) Frequency distribution of bacterial KOPS distribution in the APMV genome (AY653733.1, *L. lactis*; 5′-GAAGAAG-3′, *B. subtilis*: 5′-GAGAAGGG-3′, *E. coli*: 5′-GGGNAGGG-3′). Complementary sequences were also counted. (**b**) Density graph of the *L. lactis* type KOPS in the APMV genome (AY653733.1). KOPS on the positive and negative strand of the APMV genome were plotted separately. Red arrow: estimated *ori* region; dark gray arrows: *dif* sequence positions (**c**) Bacterial KOPS distributions in bacterial genomes (*L. lactis* Il1403: NC_002662.1, *E. coli* MG1655: NC_000913.3, *B. subtilis* 168: NC_000964.3). Red and black arrows indicate the *ori* and *ter* positions, respectively.

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
