# Peer review of "Gram-Positive Bacteria-Like DNA Binding Machineries Involved in Replication Initiation and Termination Mechanisms of Mimivirus"

_viruses, 2019, doi:10.3390/v11030267_

Round 1
Reviewer 1 Report
General comments:
Akashi and Takemura carried out bioinformatic analyzes investigating the DNA replication of the giant Mimivirus from its initiation to its termination and demonstrated a DNA binding mechanism comparable to that of certain bacteria. The results suggest that mimivirus DNA replication is initiated through the recognition of specific Chi-like quartet sequences by an helicase and involves a DNA repair machinery similar to that of Gram-positive bacteria such as Lactococcus lactis.
This article is original because few studies have focused on the mechanism of replication of Mimivirus DNA and more data are of interest to get a better insight into replication mechanims for all giant viruses. The methodology appears robust. It is interesting to show that this mechanism of DNA replication can be close to that of bacteria, which shows once again that these giant viruses have characteristics similar to that of cellular microorganisms.
Major comments:
The authors should explain that the skew GC is a tool for possibly determining the origin of replication, has been determined in bacteria and remains putative in Mimivirus.
Lines 238-240: More information should be given on the Chi sequences of the bacteria and it should be explained why that of Lactococcus lactis remains unidentified and how, under these conditions, a comparison can be made between the Chi sequences of Lactococcus lactis and Mimivirus.
Minor comments:
Line 18: "to other bacteria" instead to "other to bacteria"
Line 190: to review this sentence, it seems to miss something.
Page 8 lines 236-237: review this sentence, it seems to be missing something.
Author Response
Response to Reviewer 1 Comments
General comments:
Akashi and Takemura carried out bioinformatic analyzes investigating the DNA replication of the giant Mimivirus from its initiation to its termination and demonstrated a DNA binding mechanism comparable to that of certain bacteria. The results suggest that mimivirus DNA replication is initiated through the recognition of specific Chi-like quartet sequences by an helicase and involves a DNA repair machinery similar to that of Gram-positive bacteria such as Lactococcus lactis.
This article is original because few studies have focused on the mechanism of replication of Mimivirus DNA and more data are of interest to get a better insight into replication mechanims for all giant viruses. The methodology appears robust. It is interesting to show that this mechanism of DNA replication can be close to that of bacteria, which shows once again that these giant viruses have characteristics similar to that of cellular microorganisms.
Response:Thank you for your comments and suggestions.
Major comments:
The authors should explain that the skew GC is a tool for possibly determining the origin of replication, has been determined in bacteria and remains putative in Mimivirus.
Response:We appreciate the reviewer’s comment. Accordingly, we have added explanation about GC skew analysis at line 52. New articles were referenced, and reference numbers were revised. The revised or newly inserted sentences have been highlighted in yellow.
# line 52- > 56- (new MS)
“…segregation could be found in APMV. Plotting the nucleotide composition bias called genomic GC skew is a tool for visualizing the bias of the nucleotide composition on the genome, which is able to determine the origin of replication because the values of GC skew switches across the replication origin and its terminus [18], and which has been determined in bacteria with some improvements of the technique [19, 20, 21]. This nucleotide composition bias shaping the genomic polarity is thought to have results of the mutation and selection pressure against the different replication mechanism of leading/lagging strand [22, 23]. Although the origin of replication remains putative in Mimivirus, the GC skew analysis against the Mimivirus genome with high resolution may facilitate the detection of the signal sequence of DNA replication-initiation.Therefore, using bioinformatics, we analyzed the APMV genome to determine the DNA replication initiation/termination-segregation mechanism in detail, starting with the GC skew analysis.
# line 57- > 70- (new MS)
“GC and AT skew of the APMV genome (AY653733.1) was analyzed using a method described previously [1520, 21]. Each index was calculated using the following formulas; GC skew = [G-C]/[G+C]; AT”
18. Frank AC, Lobry JR. Asymmetric substitution patterns: A review of possible underlying mutational or selective mechanisms. Gene. 1999. doi:10.1016/S0378-1119(99)00297-8
20. Freeman JM. Patterns of Genome Organization in Bacteria. Science (80- ). 1998;279(5358):1827a-1827. doi:10.1126/science.279.5358.1827a
21. Grigoriev A. Analyzing genomes with cumulative skew diagrams. Nucleic Acids Res. 1998. doi:10.1093/nar/26.10.2286
22. Lobry JR, Sueoka N. Asymmetric directional mutation pressures in bacteria. Genome Biol. 2002;3(10):RESEARCH0058.
23. Lobry JR, Louarn J-M. Polarisation of prokaryotic chromosomes. Curr Opin Microbiol. 2003;6(2):101-108. doi:10.1016/S1369-5274(03)00024-9
Lines 238-240: More information should be given on the Chi sequences of the bacteria and it should be explained why that of Lactococcus lactis remains unidentified and how, under these conditions, a comparison can be made between the Chi sequences of Lactococcus lactis and Mimivirus.
Response:Thank you for your suggestion. We have had a mistake that the chi seq of L. lactis already determined, so we have rewritten sentences about chi sequence at line 49. After the additional analysis, the state between L. lactis and APMV was changed, therefore we have revised the manuscript as below. We have also revised the incorrect sentence at line 360.
# line 49- > 49- (new MS)
“…so the recombination is sequence specific. The sequence chi is known as a site specific recombination site that catalyzed by RecBCD pathway in E. coli, and RecB is a homologous helicase to AddA [15, 16]. Chi sequence varies among bacterial species; E. coli: 5’-GCTGGTGG-3’, B. subtilis: 5’-AGCGG-3’, L. lactis: 5’-GCGCGTG-3’[12, 15, 16, 17]. Based on…”
17. Biswas I, Maguin E, Ehrlich SD, Gruss A. A 7-base-pair sequence protects DNA from exonucleolytic degradation in Lactococcus lactis. Proc Natl Acad Sci U S A. 1995;92(6):2244-2248. doi:10.1073/pnas.92.6.2244
# line 232- > 260- (new MS)
“Each type of quartet sequence composition frequencies were calculated in APMV, threefourkinds of bacteria (B. subtilis, L. lactis, E. coli and Ricketsia prowazekii) and human mitochondrial genome (H. sapiensMT). Ricketsia prowazekiiis considered as an ancestor of mitochondria [39], while the mitogenome sequence similarity against mimivirus’s genome has been reported recently [40].These frequencies were then plotted in every pair of bacteria, mitochondria,APMV, and the oricontaining region of APMV (375k–385k) (Fig. 6). Every pair without APMV-APMV-oriand APMV-R.prowazekiiexhibited a significant difference of sequence composition (p< 0.05, Fig. 6). Furthermore, the E. coligenome exhibited the lowest p-value compared to APMV (p = 3.63e-14), and in order by B. subtilis(p = 4.112e-12) and L. lactis (p = 0.000169), indicated that L. lactisconserved the most similar quartet sequence composition of the APMV genome (Fig. 6).R.prowazekiipossesses the most similar sequence composition compared with APMV (p= 0.2528). Even if the chi sequences of L. lactisremain unidentified, our results clearly suggest that the AddA structure and corresponding chi sequence are presumably homologous between APMV and L. Lactis.The composition of R. prowazekiiand APMV was more similar than that of R. prowazekii-H. sapiensMT pair and APMV-H. sapiensMT pair (R. prowazekii-H. sapiensMT: p= 0.04685, APMV-H. sapiensMT: p= 0.00146), indicating that the APMV conserves the sequence derived from ancestor of mitochondria rather than highly evolved mitochondria in human. Interestingly, the composition of human mitochondriaand L. lactisalso showed the high similarity (p= 0.3552) rather than that of R. prowazekii-H. sapiensMT, suggesting that the mitochondrial genome still conserves the remnant of bacterial characteristics.”
39. Andersson SG, Zomorodipour A, Andersson JO, et al. The genome sequence of Rickettsia prowazekii and the origin of mitochondria. Nature. 1998;396(6707):133-140. doi:10.1038/24094
40. Seligmann H. Giant viruses as protein-coated amoeban mitochondria? Virus Res. 2018;253:77-86. doi:10.1016/j.virusres.2018.06.004
# line 360- > 395- (new MS)
“Bacillus subtilis,encodes the homolog of the RecB, AddA, and recognizes the chi sequence in E. coli(5’-AGCGG-3’)[12].”
Minor comments:
Line 18: "to other bacteria" instead to "other to bacteria"
Response:Thank you for your pointing out. We have corrected the sentence as below.
# line 18 > 18 (new MS)
“…than it wasothertobacteria.”
Line 190: to review this sentence, it seems to miss something.
Response:Thank you for your pointing out. We have changed the sentence as below.
# line 189 > 213 (new MS)
“Seven regions that were known to beconserved among AddA,and other helicases [36] were also highly conserved in the UvrD/Rep-like helicase (Fig. 4a).”
Page 8 lines 236-237: review this sentence, it seems to be missing something.
Response:Thank you for your suggestion. We have changed the sentence with additional analysis as above, the question about lines 238-240.
Reviewer 2 Report
Review of MS viruses-445700 by Akashi and Takemura
reviewer: Herve Seligmann
The ms presents bioinformatic analyses, mainly based on biases in nucleotide contents across a giant virus genome and 4 bacterial genomes, two gammaproteobacteria and two firmicutes.
These analyses detect a probable replication origin in the giant virus, viral nucleotide content patterns most similar to Lactococcus lactis as well as replication origin signals most similar to Lactococcus.
The english is understandable but would benefit from editing by a native english-speaker. I have some important comments on the contents, but overall agree with the spirit of the analyses.
My background is insufficient to evaluate the parts related to protein structures, and the parts on directions of paralogues. At least on paralogues, more explicit explanation should be given, perhaps by including a Table where positions and directions and nucleotide contents/ratios are given for paraloguous pairs, +very stepwise explanations on the performed tests.
How do you justify the choice of the 4 bacteria to which you compare the mimivirus genome, in terms of taxonomy, and molecular biology? are these the only species for which data are available?
This brings us to my most important points, related to my recently published, similar minded analyses of gene contents of "Megavirales" as compared to amoeban mitochondria, which presumably originate from rickettsia-like alphaproteobacteria (Seligmann 2018 Giant viruses as protein-coated amoeban mitochondria? Virus Res 253, 77-86 and Seligmann H 2019 Giant viruses: spore-like missing links between Rickettsia and mitochondria? Annals of the New York Academy of Sciences, in press).
The latter publication is in press, I will gladly provide the corrected galliproofs upon direct request from the authors.
The authors should for the least discuss the points made in my recent publications, to extents they resemble, and/or differ, to their results. Overall, both their and my analyses suggest bacterial-like genome structures for giant viruses.
Their quartet nucleotide composition ratio should/might be done in relation to a Rickettsia, and/or a mitochondrion. Signal sequences for replication in Rickettsia or similar bacteria should also be considered, as well as those in mitochondria, as compared to mimivirus.
For vertebrate, human mitochondria, this signal is 3'GGCCG5', from Hixson et al 1986 Both the conserved stem-loop and divergent 5'-flanking sequences are required for initiation at the human mitochondrial origin of light-strand DNA replication. J Biol Chem 264, 2384-2390.
These additional discussions and analyses would better integrate the results from this manuscript with the existing literature on the subject. Overall, at this point analyses indicate a bacteria-like genome structure, potentially a bacteria-like ancestry for giant viruses. Resolving which exact bacterial-(or mitochondrial) group(s) is of interest.
what is a quartet nucleotide composition ratio? do you mean the 264 combinations of 4 nucleotides? define how you calculated this, or explain better where this equation is in the manuscript.
Minor comments:
Kolmogorov-SmiRnov
line 134 the origin of is located... the origin of what? same problem at line 165
line 145 of these geneS
line 187 with bacterial the homologues-> rephrase, do you mean with bacterial homologues?
line 222 do accomplish this-> To accomplish this
line 226 two sequenceS
line 233 add "." after parenthesis
line 303 exact position of each sequences->sequence, delete s
line 307 delete times
line 340 suggests->suggest, delete s
line 380 of what? APMV?
Pseudomonas aeruginosa: gram- Gammaproteobacteria
Escherichia coli : Gammaproteobacteria
Bacillus subtilis : Firmicutes
Lactococcus lactis : Firmicutes
Author Response
Response to Reviewer 2 Comments
reviewer: Herve Seligmann
The ms presents bioinformatic analyses, mainly based on biases in nucleotide contents across a giant virus genome and 4 bacterial genomes, two gammaproteobacteria and two firmicutes.
These analyses detect a probable replication origin in the giant virus, viral nucleotide content patterns most similar to Lactococcus lactis as well as replication origin signals most similar to Lactococcus.
The english is understandable but would benefit from editing by a native english-speaker. I have some important comments on the contents, but overall agree with the spirit of the analyses.
Response:Thank you for your comments and suggestions.
My background is insufficient to evaluate the parts related to protein structures, and the parts on directions of paralogues. At least on paralogues, more explicit explanation should be given, perhaps by including a Table where positions and directions and nucleotide contents/ratios are given for paraloguous pairs, +very stepwise explanations on the performed tests.
Response:We appreciate your comment and agree with the point must be addressed. We have added the additional explanation about paralogue analysis as below, and we have also added the information of paralogues with locus-tag, protein accession id, sequence position, gene length, gene direction and GC content in the Supplementary Data. The corrected parts are indicated by line number and highlighted in Yellow.
# lines 72-81 > 84-103 (new MS)
“We confirmed the three kinds of paralogous gene locations found on the two GC skew shift points (296,000 bp and 882,000 bp); ankyrin repeat, serine/threonine protein kinase andcollagen triple helix repeat containing protein. The CDS start position and its direction of these types of genes annotated on the APMV genome (AY653733.1) was listed, and every pair of paralogous gene direction in each type was confirmed (matched direction: forward-forward/reverse-reverse, mismatched direction: forward-reverse/reverse-forward). This pair of directions on the physical position of the APMV genome was plotted using Circos v0.69-6 [16].First we made a gene list of the CDS information annotated as "ankyrin containing protein", "serine/threonine protein kinase" or "collagen triple helix repeat containing protein" in the APMV genome (AY653733.1), containing the data of locus tags, start/end physical positions, gene length and gene direction and GC content (Supplementary Data). Every pair of each three genes was listed and distance between these pairs from start position was calculated. This pair of directions on the physical position of the APMV genome was plotted using Circos v0.69-6 [24]. The pair of gene direction was also labeled (matched direction: forward-forward/reverse-reverse, mismatched direction: forward-reverse/reverse-forward). For drawing the graph with Circos, color coding was used to display the three kinds of genes, and states of matched/mismatched gene directions were displayed with shades of colors (ankyrin containing protein: blue, serine/threonine protein kinase: green, collagen triple helix repeat containing protein: red, matched: light, mismatched: shade). The columns of id, gene1_start_pos, gene2_start_pos, line_colour, line_thickness in the Supplementary Data without header (Supplementary Data; the pane "Paralogous gene direction") can be used for loading the data to Circos.Distance between every two CDSs in each of the three genes was plotted by the CDS direction-matching patterns on the APMV genome (“match” or “mismatch”). The statistical differences between the two groups were calculated by two-sample Kolmogorov-Smilnov (KS) test using “lawstat package” of R software with default options.”
How do you justify the choice of the 4 bacteria to which you compare the mimivirus genome, in terms of taxonomy, and molecular biology? are these the only species for which data are available?
Response:Thank you for your comments. The three bacteria in the FtsK sequence analysis (E. coli, B. suband L. lactis) have chosen based on the well-characterization of KOPS. We have added the FtsK of P. aeruginosato the analysis here because the structure of FtsK has analyzed deeply, and we needed the information about the active sites of FtsK to determine the active sites of APMV-FtsK. Accordingly, we have added the sentence as below.
# line 259- > 292- (new MS)
“…recognizes KOPS [5,26]. The DNA binding sites of FtsK has been determined in P. aeruginosa[34, 41]. We aligned the gamma domain of…”
This brings us to my most important points, related to my recently published, similar minded analyses of gene contents of "Megavirales" as compared to amoeban mitochondria, which presumably originate from rickettsia-like alphaproteobacteria (Seligmann 2018 Giant viruses as protein-coated amoeban mitochondria? Virus Res 253, 77-86 and Seligmann H 2019 Giant viruses: spore-like missing links between Rickettsia and mitochondria? Annals of the New York Academy of Sciences, in press).
The latter publication is in press, I will gladly provide the corrected galliproofs upon direct re]quest from the authors.
The authors should for the least discuss the points made in my recent publications, to extents they resemble, and/or differ, to their results. Overall, both their and my analyses suggest bacterial-like genome structures for giant viruses.
Their quartet nucleotide composition ratio should/might be done in relation to a Rickettsia, and/or a mitochondrion. Signal sequences for replication in Rickettsia or similar bacteria should also be considered, as well as those in mitochondria, as compared to mimivirus.
For vertebrate, human mitochondria, this signal is 3'GGCCG5', from Hixson et al 1986 Both the conserved stem-loop and divergent 5'-flanking sequences are required for initiation at the human mitochondrial origin of light-strand DNA replication. J Biol Chem 264, 2384-2390.
These additional discussions and analyses would better integrate the results from this manuscript with the existing literature on the subject. Overall, at this point analyses indicate a bacteria-like genome structure, potentially a bacteria- like ancestry for giant viruses. Resolving which exact bacterial-(or mitochondrial) group(s) is of interest.
Response:Thank you for your useful comments and suggestions. We really agree with those points must be addressed, so we have performed the comparative analysis of quartet nucleotide composition frequencies with human mitochondrial genome and Rickettsia prowazekiigenome, and have rewritten the result and discussions as below. The Figure 6 was replaced by a new result. The data set in the Supplementary Data was also replaced by a new one. We also thank you for your suggestion about the L-strand replication signal sequence “3'GGCCG5'” of mitochondria. The complementary of a part of this sequence “CGGC” is the most frequently found in the ori-estimated region of APMV. Therefore we have also discussed about the genome similarity among APMV, mitochondria and Rickettsia and an ancestor virus of APMV. Additional articles have been referenced in the new sentences. We have also analyzed the distribution of CGGC/GCCG sequences on the APMV genome and have added the results as Fig. 5c.
Additionally, we would like to ask you to provide your new paper in the New York Academy of Sciences, we are going to send a mail to you after this revision process.
# line 22- > 22-(new MS)
“Therefore, the replication initiation and termination segregation systems of APMV are presumably mediated by DNA repair machineries similar to that of gram-positive bacteria such as L. lactis.”
# line 106- > 128- (new MS)
“Every quartet sequence compositions on APMV and the bacterial genomes were confirmed with the compseq program of EMBOSS 6.6.0.0 [31] with “-word 4 ” option, using the following genome data: APMV: AY653733.1; L. lactisIl1403: NC_002662.1; E. coliMG1655: NC_000913.3; B. subtilis168: NC_000964.3;R. prowazekii: NC_000963.1; H. sapiensmitochondria (MT): CM001971.1.”
# line 223- > 246- (new MS)
“The four types of sequences that had a ratio greater than 2 contained 75-100% of GC nucleotides, and two of four sequences, CGGC and CCGC, exhibited significantly different ratios from the other sequences (p< 0.05, Fig. 5a). Further, two sequence (CCGC and GCGG) were complementary to each other (Fig. 5a). This pair of sequences (CCGC and GCGG) was identical to four out of five nucleotides on the chi sequence, which was recognized by Bacillus subtilis-AddA (5’-AGCGG-3’) [12]. Additionaly, the sequence CGGC is known to be a part of complementary sequence that is known as a replication signal sequence of mitochondrial genome in human, 3’-GGCCG-5’ [38].Therefore, we analyzed the distribution of thesequartet sequence in the APMV genome.”
38. Hixson JE, Wong TW, Clayton DA. Both the conserved stem-loop and divergent 5’-flanking sequences are required for initiation at the human mitochondrial origin of light-strand DNA replication. J Biol Chem. 1986;261(5):2384-2390.
# line 229- > 254- (new MS)
“The sequence densities of CCGC and GCGG in the APMV genome showed that both sequences had two peaks, and one of each peak was located on the ori-containing region, while the others were on the axial symmetric position (about 800,000 bp, Fig. 5b). Similar to this, those of CGGC and GCCG exhibited two peaks and the peak on the 5’ side in GCCG located on the ori-containing region, although that of the CGGC slightly skewed from the ori-containing region to the 5’ end of the genome (Fig. 5c).”
# line 232- > 260- (new MS)
“Each type of quartet sequence composition frequencies were calculated in APMV, threefourkinds of bacteria (B. subtilis, L. lactis, E. coli and Ricketsia prowazekii), and human mitochondrial genome (H. sapiensMT). Ricketsia prowazekiiis considered as an ancestor of mitochondria [39], while the mitogenome sequence similarity against mimivirus’s genome has been reported recently [40].These frequencies were then plotted in every pair of bacteria, mitochondria,APMV, and the oricontaining region of APMV (375k–385k) (Fig. 6). Every pair without APMV-APMV-oriand APMV-R.prowazekiiexhibited a significant difference of sequence composition (p< 0.05, Fig. 6). Furthermore, the E. coligenome exhibited the lowest p-value compared to APMV (p = 3.63e-14), and in order by B. subtilis(p = 4.112e-12) and L. lactis (p = 0.000169), indicated that L. lactisconserved the most similar quartet sequence composition of the APMV genome (Fig. 6).R.prowazekiipossesses the most similar sequence composition compared with APMV (p = 0.2528). Even if the chi sequences of L. lactisremain unidentified, our results clearly suggest that the AddA structure and corresponding chi sequence are presumably homologous between APMV and L. Lactis.The composition similarity of R. prowazekii-APMV pair was more similar than that of R. prowazekii-H. sapiensMT pair and APMV-H. sapiensMT pair (R. prowazekii-H. sapiensMT: p= 0.04685, APMV-H. sapiensMT: p= 0.00146), suggesting that the APMV conserves the sequence derived from ancestor of mitochondria rather than highly evolved mitochondria in human.Interestingly,the composition of human mitochondriaand L. lactisalso showed the high similarity (p= 0.3552) rather than that of R. prowazekii-H. sapiensMT, suggesting that the mitochondrial genome still conserves the remnant of bacterial characteristics.”
39.Andersson SG, Zomorodipour A, Andersson JO, et al. The genome sequence of Rickettsia prowazekii and the origin of mitochondria. Nature. 1998;396(6707):133-140. doi:10.1038/24094
40.Seligmann H. Giant viruses as protein-coated amoeban mitochondria? Virus Res. 2018;253:77-86. doi:10.1016/j.virusres.2018.06.004
# line 246- > 280- (new MS)
“(b and c) Density graph of the 5’-CCGC-3’ sequence and its complementary sequence 5’-GCGG-3’5’-CCGC-3’/5’-GCGG-3’ and 5’-CGGC-3’/5’-GCCG-3’on the APMV genome (AY653733.1). Red arrowhead on the scale indicates the estimated location of the ori-region (382,000 ± 5,000 bp).”
# line 250- > 283- (new MS)
“Figure 6. Comparative analysis of quartet nucleotide composition frequencies among Acanthamoeba polyphaga mimivirus(APMV), human mitochondria (H. sapiensMT)and threefourbacterial genomes. Pairs of every frequency between two species were plotted with an approximate line. “APMV ori” indicates the frequencies of sequence compositions in the estimated location of the ori-region (375–385 kb region of AY653733.1). P-values on each top left corner of the graphs are the significant differences calculated between two groups as determined by KS-test.”
# line 372- > 406- (new MS)
“Phosphate binding sites were not conserved in UvrD/Rep-like helicase of APMV at the sequence level, but these regions contain polar amino acids similar to B. subtilisAddA (Fig. 4b), which indicates that the UvrD/Rep-like helicase of APMV could also bind phosphate at the 3’ end of the chi sequence. Furthermore, the other quartet sequence 5’-CGGC-3’, which was detected on the ori-estimated region of APMV(Fig.5a and 5c), is reported to be a part of DNA replication signal sequence of mitochondria (3’-GGCCG-5’) [38], indicating that the DNA replication of APMV is also initiated by host replication machineries involved in the replication of mitochondrial DNA.”
38. Hixson JE, Wong TW, Clayton DA. Both the conserved stem-loop and divergent 5’-flanking sequences are required for initiation at the human mitochondrial origin of light-strand DNA replication. J Biol Chem. 1986;261(5):2384-2390.
#Fig. 5 ver.2
(Revised figure is in the PDF file.)
#Fig. 6 ver.2
(Revised figure is in the PDF file.)
# line 387- > 425- (new MS)
“These results suggest that APMV-FtsK might be functionally homologous to that of bacteria, corresponding to the results that the sequence composition of the APMV genome was more closely related to L. lactis, rather than B. subtilisand E. coli(Fig. 6).”
# line 390- > 427- (new MS)
“…Therefore, the mechanisms of interaction between these proteins and specific sequences are presumably homologous between APMV and bacteria such as L. lactis, although the further molecular biological studies and structural analyses are required to certify this model.
The phylogenetic relations between Rickettsiaand mitochondria and between mitochondria and Mimivirus were described previously [39, 40]. Interestingly, the genome size of Rickettsia is 1.1Mb and its GC content is 29% [39], those of which are highly similar to APMV (1.2Mb, 28%). Furthermore, the backbone of Mimivirus genome is reported to be derived from the ancestor of mitochondria [40]. Considering the quartet sequence similarities (Fig. 6) and the phylogenetic relation between Rickettsiaand mitochondria, the ancestor of Mimivirus had infected to the ancestor of Eukaryotic cells (LACA: last archaeal common ancestor) before the endosymbiosis of mitochondrial ancestor to the FECA (the first eukaryotic common ancestor), while the ancestor of mitochondria and Rickettsiahad also infected to the ancestor of Eukaryotic cells. This ancestral virus presumably harbored shorter genomic DNA than that of the nowadays Mimivirus, and it acquired the long genome from ancestor of mitochondria by genomic fusion. The sequence 5’-CGGC-3’ found in APMV-oriregion might also acquired from the ancestor of mitochondria, which is still conserved as a DNA replication signal sequence (3’-GGCCG-5’)in the mitochondrial genome [38]. Moreover, the co-infection to (or to be preyed on) the LACA cells might have occurred not only in ancestor of mitochondria and Mimivirus but also in other bacteria, therefore APMV conserves the bacteria-like machineries such as UvrD/Rep-like helicase and APMV-FtsK derived from the co-infected ancestor of gram-positive bacteria by horizontal gene transfer.”
38. Hixson JE, Wong TW, Clayton DA. Both the conserved stem-loop and divergent 5’-flanking sequences are required for initiation at the human mitochondrial origin of light-strand DNA replication. J Biol Chem. 1986;261(5):2384-2390.
39.Andersson SG, Zomorodipour A, Andersson JO, et al. The genome sequence of Rickettsia prowazekii and the origin of mitochondria. Nature. 1998;396(6707):133-140. doi:10.1038/24094
40.Seligmann H. Giant viruses as protein-coated amoeban mitochondria? Virus Res. 2018;253:77-86. doi:10.1016/j.virusres.2018.06.004
# line 395- > 448- (new MS)
“Here we show a proposed model of initiation and termination of DNA replication and chromosome segregation for APMV. The estimated ori-region exists at the 382 kb position in the genome, which contains the chi-like sequence recognized by B. subtilis AddA, which is homologous to the UvrD/Rep-like helicase of APMV. The other sequence has a homology of a DNA replication signal sequence of mitochondria, indicating that the DNA replication of APMV may initiate with the replication machineries of mitochondria.The KOPS distribution pattern and the structure of APMV-FtsK indicate that the KOPS recognition system by APMV-FtsK is similar to that of L. lactis. Consequently, replication initiation and termination segregation systems of APMV are presumably mediated by DNA repair machineries, similar to that of gram-positive bacteria, such as L. lactis. Overall, our findings show some similarity betweenL. lactisand the APMV genome. APMV has an AT-rich genome (GC%: 28%) and L. lactishas conserved a low GC genome (GC%: 35%), suggesting that the genome maintenance mechanism of APMV is similar to L. lactis.Furthermore, the comparison of quartet sequence compositions show the similarity between APMV and Rickettsia,which may have the most closest common ancestor of mitochondria, indicating that the Mimivirus have acquired the large bacteria-like genome and their DNA replication machineries from ancestor of mitochondria during the co-infection to the LACA cells. The evolutionary history of APMV remains unclear; however, the further analyses of such a chimeric genome of APMV may illustrate the early stage of evolution of eukaryotic cells and Mimivirus.the link between APMV and gram-positive bacteria hints at solutions that may solve this mystery in the future.”
what is a quartet nucleotide composition ratio? do you mean the 264 combinations of 4 nucleotides? define how you calculated this, or explain better where this equation is in the manuscript.
Response:Thank you for your suggestion. The word “quartet nucleotide composition ratio” was missing in the Materials and Methods. We have added the following sentence.
# line 109- > 131- (new MS)
“To compare the compositions between the ori-estimated region and the whole genome of APMV, sequence composition was also confirmed on the 375–385 kb region, and the ratio of ori-region per whole genome (quartet nucleotide composition ratio)was calculated as follows: ratioquartet nucleotide composition ratio= [ori Observed/Expected Frequency]/[all genome Observed/Expected Frequency].”
# line 221- > 244- (new MS)
“Subsequently, we sought to find the sequence that was related to the origin of recognition on the probable ori region. Toaccomplish this, we first calculated the quartet nucleotide composition ratio of the ori-containing region per total genome (Fig. 5). For calculation of the ratio, see Materials and Methods.”
Minor comments:
Kolmogorov-SmiRnov
Response:Thank you for your pointing out. We have corrected wrong words in the sections 2.2, 2.3, 2.6 of Materials and Methods.
# lines 68, 80, 116 > 81, 102, 140 (new MS)
“…by two-sample Kolmogorov-SmiRnov (KS) test…”
line 134 the origin of is located... the origin of what? same problem at line 165
Response:Thank you for your pointing out. We have corrected the sentence.
# lines 134 > 157 (new MS)
“suggesting that the origin of DNA replicationis located in this region (Fig. 1).”
line 145 of these geneS
Response:Thank you for your pointing out. We have corrected the word.
# lines 145 > 168 (new MS)
“Paralogues of these geneswere…”
line 187 with bacterial the homologues-> rephrase, do you mean with bacterial homologues?
Response:Thank you for your suggestion. Accordingly, we have changed the sentence.
# lines 187 > 210 (new MS)
“…with thebacterial thehomologues; …”
line 222 do accomplish this-> To accomplish this
Response:Thank you for your pointing out. We have corrected the sentence.
# lines 222 > 245 (new MS)
“DoToaccomplish this,…”
line 226 two sequenceS
Response:Thank you for your pointing out. We have corrected the word.
# lines 226 > 249 (new MS)
“two sequences(CCGC and GCGG)”
line 233 add "." after parenthesis
Response:Thank you for your pointing out. We have corrected.
# lines 233 > 261 (new MS)
“…L. lactisand E. coli and Ricketsia prowazekii) and human mitochondrial genome (H. sapiensMT).”
line 303 exact position of each sequences->sequence, delete s
Response:Thank you for your pointing out. We have corrected.
# lines 303 > 337 (new MS)
“…of each sequences.”
line 307 delete times
Response:Thank you for your pointing out. We have corrected.
# lines 98, 219, 292, 307 > 120, 242, 326, 341 (new MS)
“…1,000 times”
line 340 suggests->suggest, delete s
Response:Thank you for your pointing out. We have corrected the word.
# lines 340 > 374 (new MS)
“…GC skew plot suggeststhat…”
line 380 of what? APMV?
Response:Thank you for your pointing out. We have corrected the sentence.
# lines 380 > 417 (new MS)
“…L. lactis-KOPS ofwas similar…”

Round 2
Reviewer 2 Report
My major concern with this manuscript is the language. Please get help from native english speakers. Cut long sentences into shorter ones. The science is really nice, so it is worth the effort to better the language.
Two points remain:
the abstract does not stress enough the new findings on mimivirus-mitochondrion-rickettsia similarities.
I just published an additional paper on mitochondria-virus similarities which you might want to cite :
Seligmann H 2019 Giant viruses: spore-like missing links between Rickettsia and mitochondria? Ann NY Acad Sci, in press. The galleyproof should be available on researchgate.
the test is called Kolmogorov-Smirnov
I wrote SmiRnov to stress the difference with your incorrect spellin "Smilnov", but the "r" is a minuscule, not a majuscule, sorry for causing the confusion.
Author Response
Response to Reviewer 2 Comments
reviewer: Herve Seligmann
My major concern with this manuscript is the language. Please get help from native english speakers. Cut long sentences into shorter ones. The science is really nice, so it is worth the effort to better the language.
Response:Thank you for your comments and suggestions.
We would like to assign the language-editing system in MDPI to this manuscript, after the revision process.
Two points remain:
the abstract does not stress enough the new findings on mimivirus- mitochondrion-rickettsia similarities.
Response:Thank you for your pointing out. We have changed the abstract. Additionally, we have added key words “mitochondria”, “Rickettsia”, “Gram-positive bacteria” and “eukaryogenesis”, while removing the key word “dif” for reflecting the contents.
# lines 10- (new MS)
Detailed mechanisms of replication-initiation and termination-segregation events were not yet known in Acanthamoeba polyphaga mimivirus(APMV). Here we show detailed bioinformatics-based analyses of DNAchromosomal-replication in APMV from initiation to termination mediated by proteins bound to specific DNA sequences. Using GC/AT skew and coding sequence skew analysis, we estimated that the origin of DNAreplication-originis located at 382 kb in the APMV genome. We performed homology-modeling analysis of APMV-FtsK gamma domain related to FtsK-orienting polar sequences (KOPS) binding, suggesting that there was anunique insertion in the gamma domain, which maintains the structure of the DNA binding motif. The distribution pattern of KOPS in the APMV genome was more similar to Lactococcus lactisthan other bacteria.Further, UvrD/Rep-like helicase in APMV was homologous to Bacillus subtilis-AddA, while the chi-like quartet sequence 5’-CCGC-3’was frequently found in the ori-estimated region rather than throughout the genome. This suggests, suggesting that APMVchromosomal-replication of APMVis initiated via chi-like quartetsequence recognition by UvrD/Rep-like helicase. Therefore, the replication-initiation and termination-segregation systemsof APMV are presumably mediated by DNA repair-machineries similar to that ofderived fromgram-positive bacteria. Moreover, the other frequently appeared quartet sequence 5’-CGGC-3’ in the ori-region was homologous to the mitochondrial-signal sequence of replication-initiation, while the comparison of quartet sequence-composition in APMV/Rickettsia-genome showed significantly similar values, suggesting that APMV also conserves the mitochondrial replication-system which have acquired from an ancestral genome of mitochondria during eukaryogenesis.
# lines 28 (new MS)
Keywords: DNA replication; ori; dif;Mitochondria; Rickettsia; Gram-positive-bacteria; APMV; Mimivirus; Giant virus; Eukaryogenesis.
I just published an additional paper on mitochondria-virus similarities which you might want to cite :
Seligmann H 2019 Giant viruses: spore-like missing links between Rickettsia and mitochondria? Ann NY Acad Sci, in press. The galleyproof should be available on researchgate.
Response:Thank you for kindly sending us your new paper. We have cited this in our manuscript and have added the new sentences as below (the last part of Discussion section).
# lines 431- (new MS)
The phylogenetic relations between Rickettsiaand mitochondria and between mitochondria and Mimivirus were described previously [39, 40]. Interestingly, the genome size of Rickettsiais 1.1Mb and its GC content is 29% [39], those of which are highly similar to APMV (1.2Mb, 28%). Furthermore, the backbone of Mimivirus genome is reported to be derived from the ancestor of mitochondria [40]. Considering the quartet sequence similarities (Fig. 6) and the phylogenetic relation between Rickettsiaand mitochondria, the ancestor of Mimivirus had infected to the ancestor of Eukaryotic cells (LACA: last archaeal common ancestor) before the endosymbiosis of mitochondrial ancestor to the FECA (the first eukaryotic common ancestor), while the ancestor of mitochondria and Rickettsiahad also infected to the ancestor of Eukaryotic cells. This ancestral virus presumably harbored shorter genomic DNA than that of the nowadays Mimivirus, and it acquired the long genome from ancestor of mitochondria by genomic fusion. The sequence 5’-CGGC-3’ found in APMV-oriregion might also beacquired from the ancestor of mitochondria, which is still conserved as a DNA replication signal sequence (3’-GGCCG-5’) in the mitochondrial genome [38]. It has been reported that the comparison between mitochondrial genes and Rickettsiagenome shows much higher similarities than that between mito-genes and megaviral genome(poxvirus), while the synteny of these three species are significantly conserved [45]. Thus, according to our results, the genomic remnants of the ancestor of mitochondria may be still conserved in Mimivirus genome more than poxvirus one.Moreover, the co-infection to (or to be preyed on) the LACA cells might have occurred not only in ancestor of mitochondria and Mimivirus but also in other bacteria, therefore APMV conserves the bacteria-like machineries such as UvrD/Rep-like helicase and APMV-FtsK derived from the co-infected ancestor of gram-positive bacteria by horizontal gene transfer.
45. Seligmann H. Giant viruses: spore-like missing links between Rickettsia and mitochondria?Ann. N.Y. Acad. Sci. 2019. doi: 10.1111/nyas.14022
the test is called Kolmogorov-Smirnov
I wrote SmiRnov to stress the difference with your incorrect spellin "Smilnov", but the "r" is a minuscule, not a majuscule, sorry for causing the confusion.
Response:Thank you for your pointing out. We have misunderstood that, so we have corrected the sentences.
# lines 84, 105, 142 (new MS)
“…by two-sample Kolmogorov-Smirnov (KS) test…”